# OpenReview forum: "$\textit{Who Speaks Matters}$: Analysing the Influence of the Speaker’s Ethnicity on Hate Classification"
_NeurIPS.cc/2024/Workshop/SafeGenAi — SafeGenAi Oral_

### Official Review · Reviewer_GQjf · 2024-10-08
**Review on Who Speaks Matter**

**Rating:** 7
**Confidence:** 4

**Review:**

The authors investigate how explicit/implicit ethnic markers impact LLMs' ability to detect hate speech. This research can help uncover hidden biases in LLMs and help improve hate speech detection.

## Pros
- The methodology clearly states the dataset specifications and the source used.
- The overall flow/structure of the paper is smooth.
- The authors explicitly state the limitations faced.

## Questions/feedback about the paper
- Some figures need revision. For example, figure 1 does not need the speaker identity column. Table 2 is unnecessary (or could be converted into a paragraph). Figure 2 could be clearer in the direction of the flips: writing "NH to H" instead of just "NH" clears up the direction of the flip. Figure 3's bottom row is nearly identical, and the font is quite small to read unless you zoom in significantly. I recommend moving some of Figure 3's heatmaps to the appendix and only focusing on what the authors want to emphasize.
- An example of the Jaccard similarity on some sample texts would strengthen the claim on lines 125-126.

Overall, the paper demonstrates some interesting insights on racial/ethnic biases within LLMs. However, improving the figures would make the paper even stronger.

---

### Official Review · Reviewer_mR72 · 2024-10-10
**Novel idea and well implemented.**

**Rating:** 9
**Confidence:** 3

**Review:**

This paper investigates how Large Language Models (LLMs) used for hate speech detection change their classification when explicit or implicit markers of the speaker's ethnicity are injected into the input. The study focuses on five ethnicities: British, Indian, Singaporean, Jamaican, and African-American, and it reveals that implicit dialect markers tend to cause more classification flips than explicit markers. The paper emphasizes that larger models are more robust, but the results underscore the brittleness of LLMs in handling dialect variations, which is crucial for ensuring fairness in content moderation.

Pros:
- The paper addresses an important issue—bias in LLM-based hate speech detection, especially as these models are increasingly used in content moderation.
- It explores the novel idea of investigating both explicit and implicit (dialectal) markers of ethnicity and their effect on model behavior.
- Clear methodology that includes perturbing hate speech inputs with explicit identity markers.

Cons:
- While the analysis of four models is useful, the paper could have explored other models advertised as safer or bias-reduced (e.g., Claude). A scaling or capabilities analysis (do bigger models / more capable models get better or worse at certain types of ethnic bias?) would be interesting too.